# An intrinsic cell cycle timer terminates limb bud outgrowth

Joseph Pickering[1], Constance A Rich[1], Holly Stainton[1], Cristina Aceituno[2], Kavitha Chinnaiya[1], Patricia Saiz-Lopez[2], Marian A Ros[2,3], Matthew Towers[1]*

[1]Department of Biomedical Science, The Bateson Centre, University of Sheffield, Sheffield, United Kingdom; [2]Instituto de Biomedicina y Biotecnología de Cantabria, IBBTEC (CSIC-Universidad de Cantabria), Santander, Spain; [3]Departamento de Anatomía y Biología Celular, Facultad de Medicina, Universidad de Cantabria, Santander, Spain

**Abstract** The longstanding view of how proliferative outgrowth terminates following the patterning phase of limb development involves the breakdown of reciprocal extrinsic signalling between the distal mesenchyme and the overlying epithelium (e-m signalling). However, by grafting distal mesenchyme cells from late stage chick wing buds to the epithelial environment of younger wing buds, we show that this mechanism is not required. RNA sequencing reveals that distal mesenchyme cells complete proliferative outgrowth by an intrinsic cell cycle timer in the presence of e-m signalling. In this process, e-m signalling is required permissively to allow the intrinsic cell cycle timer to run its course. We provide evidence that a temporal switch from BMP antagonism to BMP signalling controls the intrinsic cell cycle timer during limb outgrowth. Our findings have general implications for other patterning systems in which extrinsic signals and intrinsic timers are integrated.

DOI: https://doi.org/10.7554/eLife.37429.001

*For correspondence:
m.towers@sheffield.ac.uk

**Competing interests:** The authors declare that no competing interests exist.

## Introduction

Detailed models of pattern formation often involve cells being informed of their position in the embryo by interpreting gradients of extrinsic signals, or by measuring intrinsic time. Recently, it has become evident that both processes can operate together - either in parallel, such as in antero-posterior patterning of the limb (thumb to little finger) (*Harfe et al., 2004*; *Towers et al., 2011*) and in dorso-ventral patterning of the neural tube (*Dessaud et al., 2007*) - or sequentially, such as in proximo-distal patterning of the limb (shoulder to finger tips) (*Cooper et al., 2011*; *Roselló-Díez et al., 2014*; *Saiz-Lopez et al., 2015*; *Saiz-Lopez et al., 2017*). Thus, understanding the precise requirement for extrinsic signals and intrinsic time in pattern formation is a challenging task. In addition, little is known about how these mechanisms operate together to terminate growth following pattern formation.

During early stages of proximo-distal patterning of the chick wing (stage Hamburger Hamilton HH19 – see Materials and methods for embryonic days), signals from the main body of the embryo specify proximal identity (humerus) (*Cooper et al., 2011*; *Mercader et al., 1999*; *Mercader et al., 2000*; *Roselló-Díez et al., 2011*). Once distal mesenchyme cells of the wing bud are displaced by growth away from proximal signals, there is a switch to an intrinsic timing mechanism that specifies distal identity (elbow to digits) (*Saiz-Lopez et al., 2015*). This intrinsic timer involves cells executing a programme of proliferation, expressing regulators of distal position (i.e. *Hoxa13*) and altering their cell surface properties, which are considered to encode position along the proximo-distal axis. In addition, distal limb bud mesenchyme cells produce a factor that maintains the overlying apical ectodermal ridge (*Zwilling and Hansborough, 1956*) - a thickening of the distal epithelium, which is

required for bud outgrowth until HH29/30 (*Saiz-Lopez et al., 2015*; *Saunders, 1948*; *Summerbell, 1974*). This factor is the Bone Morphogenetic Protein (BMP) antagonist Gremlin1 (Grem1), which maintains expression of genes encoding Fibroblast Growth Factors (FGFs) in the apical ectodermal ridge until HH27 (*Merino et al., 1999*; *Pizette and Niswander, 1999*; *Zúñiga et al., 1999*). Reciprocal signalling between the apical ectodermal ridge and the underlying mesenchyme is considered to maintain outgrowth, first by repressing apoptosis, and then by permissively maintaining cell proliferation (*Dudley et al., 2002*; *Saiz-Lopez et al., 2015*). Thus, the breakdown of the epithelial-mesenchyme (e-m signalling) feedback loop at HH27, and the subsequent loss of apical ectodermal ridge signalling, is considered necessary to terminate limb bud outgrowth at the end of the patterning phase (*Zwilling and Hansborough, 1956*; *Scherz et al., 2004*; *Verheyden and Sun, 2008*). Note that the final distal phalanx is laid down at around HH28 in the chick wing (*Saunders, 1948*; *Summerbell, 1974*). This could suggest that a switch from an intrinsic timer in the mesenchyme back to extrinsic signalling by the apical ectodermal ridge occurs at later stages of proximo-distal patterning to terminate outgrowth. Later growth of the limb after the apical ectodermal ridge has regressed is caused by the expansion of osteogenic progenitor cells in the developing long bones.

However, we recently showed that grafts of HH27 mesenchyme cells expressing Green Fluorescent Protein (GFP) that were made to younger HH20 buds, and which were left to develop for 24 hr until HH24 (now referred to as HH24 graft or HH24g), completed their proliferation programme in the presence of an overlying host apical ectodermal ridge (*Saiz-Lopez et al., 2017*) red lines - *Figure 1a*, compare similar trajectories between HH27 to HH29 of normal development – black lines). Thus, HH24g mesenchyme (distal 150 microns of graft) has a similar cell cycle profile to HH29 mesenchyme (stage of donor limb), rather than contralateral HH24 mesenchyme (stage of host – *Figure 1a* - red lines). Although wing buds that develop with these grafts produce the three main segments of the wing, the autopod (digits and wrist) is very stunted due to the reduced growth of the grafted tissue relative to host tissue (*Saiz-Lopez et al., 2017*). Therefore, an interpretation of these findings is that proliferative outgrowth terminates intrinsically in the distal mesenchyme (curved arrows in graft in green, *Figure 1b*) independently of e-m signalling (arrows between ridge and mesenchyme, *Figure 1b*). An alternative possibility is that extrinsic e-m signalling had irreversibly broken down in donor tissue at the time of grafting and causes loss of proliferation in the graft (arrows absent in HH27 wing in *Figure 1c*).

In this study, we show that proliferative outgrowth at the end of the patterning phase terminates intrinsically in the distal mesenchyme of the chick wing bud in the presence of e-m signalling. Our data provide evidence that an intrinsic switch from autocrine BMP antagonism to autocrine BMP signalling controls the cell cycle programme in distal mesenchyme during limb bud outgrowth.

## Results

### RNA sequencing reveals reversible and stable gene expression

To gain insights into how intrinsic cell cycle timing is maintained in limb development, we used RNA-seq to determine how the transcriptome changes over time in the distal part of the wing bud during normal development and then compared this to the HH24g transcriptome. To achieve this, we performed RNA-seq on pooled blocks of distal cells from HH24, HH24g, HH27 and HH29 chick wing buds (three replicates of 12 blocks taken from distal tissue excluding the polarizing region– note one HH24 sample failed quality control - see Materials and methods). The region of undifferentiated mesenchyme is considered to extend proximally beneath the apical ectodermal ridge by 200 – 300 microns and to remain at a consistent length throughout outgrowth (*Summerbell et al., 1973*), but for our analyses, we used RNA extracted from the distal 150 microns of control buds and grafts to ensure consistency between all samples. We compared HH24g with contralateral HH24 datasets and HH24g with HH29 datasets (the stage the graft would have developed to if left in situ in the donor wing). This identified 154 genes that are differentially expressed (>2 fold difference at a p-value of <0.0005) between the two comparisons: 55 genes between the HH24g-HH24 pair and 99 genes between the HH24g-HH29 pair (*Supplementary files 1* and *2*). Pearson correlation of the normalised data reveals that overall gene expression levels in HH24g distal cells are closer to HH24 than to HH29 cells, suggesting a good degree of resetting behaviour (*Figure 1d*). This is unexpected

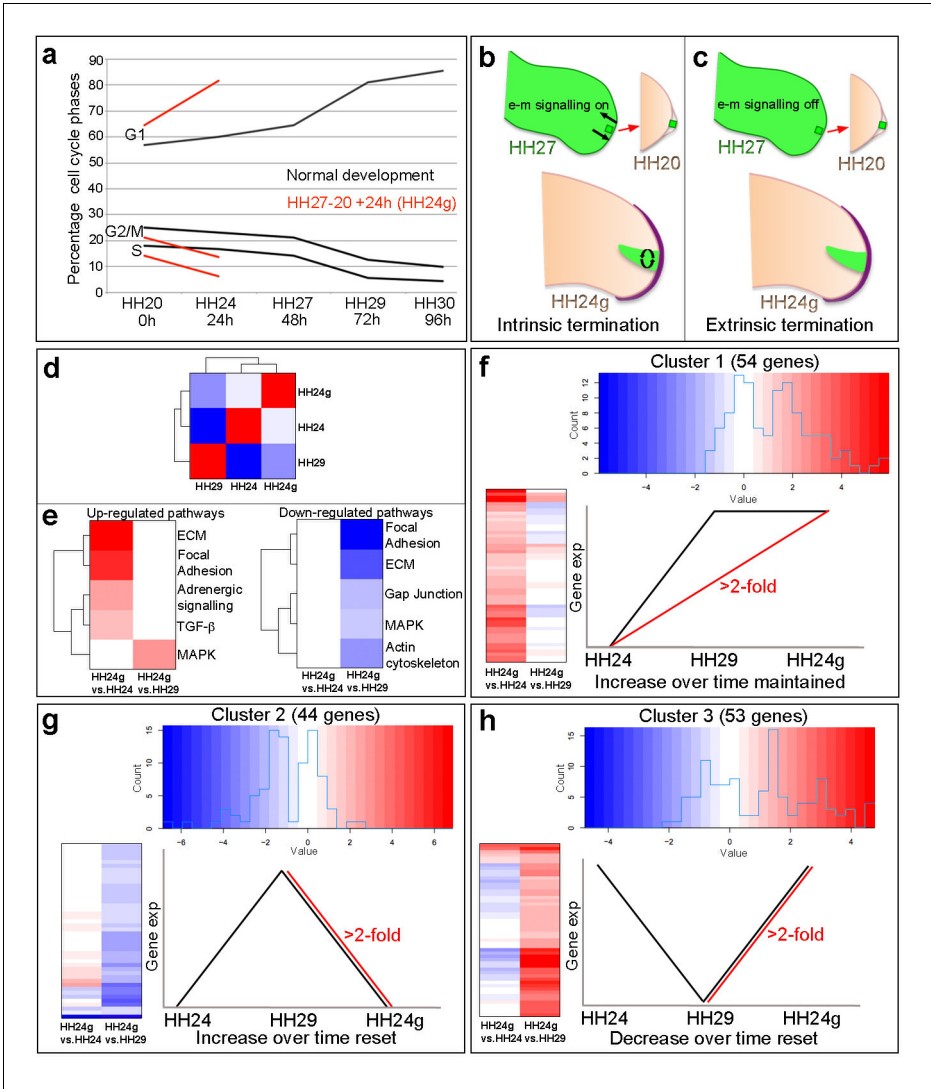

**Figure 1.** Cell cycle and RNA-seq analyses of chick wing distal tips. (a) Decline in cell cycle rate determined by proportions of cells in G1-, S- and G2/M-phases between HH20 and HH30 in the distal chick wing bud (black lines *Saiz-Lopez et al., 2017*). Red lines show maintenance of cell cycle program in grafts of HH27 distal tips made to HH20 wing buds left for 24 hr until HH24 (HH24g - tissue would have progressed from HH27 to HH29 in donor) – note trajectories follow same pattern as black lines between HH27 and HH29. (b–c) Procedure for making HH27-20 grafts (*Saiz-Lopez et al., 2017*) and predictions for loss of proliferative growth in HH24g mesenchyme. (b) Intrinsic termination: e-m signalling (arrows in HH27 wing bud) maintained between mesenchyme and apical ridge but proliferation declines intrinsically in mesenchyme independently of e-m signalling (curved arrows in HH24g mesenchyme). (c) Extrinsic termination: e-m signalling irreversibly lost between graft and apical ridge in HH27 bud (arrows absent) and proliferation lost in graft. (d) Heat-map showing the correlation (Pearson) of the normalised RNA-seq data collapsed to the mean expression per group and the degree of correlation indicated by the colour (red: higher, blue: lower). (e) KEGG analyses across pairwise contrasts with degree in pathway change indicated by the colour (red: up-regulated, blue: down-regulated). (f–h) Clustering of RNA-seq data across pairwise contrasts with degree of gene expression change indicated by the colour (red: higher, blue: lower). (f) Cluster 1: genes that increase between HH24 and HH29 (black line) and are maintained in HH24g (red line - > 2 fold higher in HH24g than HH24). (g) Cluster 2: genes that increase between HH24 and HH29 (black line) and reset in HH24g (red line - > 2 fold lower in HH24g than HH29). (h) Cluster 3: genes that decrease between HH24 and HH29 (black line) and reset in HH24g (red line - > 2 fold higher in HH24g than HH29).

DOI: https://doi.org/10.7554/eLife.37429.002

The following figure supplements are available for figure 1:

*Figure 1 continued*

**Figure supplement 1.** Cluster 1 Clustering of RNA-seq data across pairwise contrasts with degree of gene expression change indicated by the colour (red: higher, blue: lower).

DOI: https://doi.org/10.7554/eLife.37429.003

**Figure supplement 2.** Cluster 2 Clustering of RNA-seq data across pairwise contrasts with degree of gene expression change indicated by the colour (red: higher, blue: lower).

DOI: https://doi.org/10.7554/eLife.37429.004

**Figure supplement 3.** Cluster 3 Clustering of RNA-seq data across pairwise contrasts with degree of gene expression change indicated by the colour (red: higher, blue: lower).

DOI: https://doi.org/10.7554/eLife.37429.005

considering the idea that older mesenchyme cells maintain both their proliferative and patterning potential when grafted to a younger bud (*Saiz-Lopez et al., 2017*).

We performed KEGG (Kyoto Encyclopaedia of Genes and Genomes) analyses to identify up- and down-regulated pathways between the two pairwise comparisons (*Figure 1e*, *Supplementary file 1* and *2* - the pathways identified were those that contained two or more genes at a >2 fold difference at a p-value of 0.05 – it should be noted that this analysis is limited since only approximately 20% of chicken genes belong to verified KEGG pathways). There are five down-regulated pathways identified by the KEGG analyses in HH24g compared to HH29 distal cells - focal adhesion, ECM (extracellular matrix), gap junction, MAPK (Mitogen-Activated Protein Kinase) and actin cytoskeleton - suggesting a general reset of genes in these pathways that increase in expression over time between HH24 and HH29. In addition, there are four up-regulated pathways identified by the KEGG analyses in HH24g compared to HH24 distal cells - ECM, focal adhesion, adrenergic signalling, TGFβ Transforming Growth Factor Beta) - suggesting maintained expression of genes in these pathways that generally increase in expression over time. In addition, the MAPK pathway is also up-regulated in HH24g compared to HH29 distal cells, showing the complex nature of this signalling pathway, and suggesting some genes in this pathway that decrease in expression over time can be reset. Interestingly, pathways represented by genes whose decrease in expression over time is maintained in HH24g distal cells were not found.

To identify genes that share similar behaviour across the two pairwise contrasts, we performed hierarchical clustering analyses of the 154 genes significantly altered in expression between the HH24g/HH24 and HH24g/HH29 datasets. This divided the genes into three clusters (See Materials and methods – note three genes were excluded that fell into more than one cluster). Cluster 1 (*Figure 1f*, *Figure 1—figure supplement 1*, *Supplementary file 3*) contains 54 genes that increase in expression between HH24 and HH29 during normal development (black line) and are maintained in HH24g distal cells (maintain a >2 fold increase in expression compared to equivalent area in contralateral HH24 buds – red line). Cluster 2 (*Figure 1g*, *Figure 1—figure supplement 2* and *Supplementary file 4*) contains 44 genes that normally increase in expression between HH24 and HH29 (black line) and are not maintained in HH24g distal cells (>2 fold decrease in expression compared to HH29 buds - red line). Cluster 3 (*Figure 1h*, *Figure 1—figure supplement 3* and *Supplementary file 5*) contains 53 genes that decrease in expression between HH24 and HH29 (black line) and are reset in HH24g grafts (>2 fold increase in expression compared to HH29 buds - red line). Thus, genes in cluster one reflect stability of expression while genes in clusters 2 and 3 show reversibility of expression. In consensus with the KEGG analyses, a cluster containing genes whose decrease in expression over time is maintained in HH24g distal cells was not found, indicating that the young environment potently reactivates gene expression. These data suggest that many genes that are preferentially expressed during early stages of wing development can be reset to some degree (Cluster 3). However, later expressed genes show differential resetting behaviour in an earlier environment (Clusters 1 and 2).

## Bmp2/4/7 expression is maintained in HH24g mesenchyme

To identify signalling pathways associated with the maintenance of cell cycle timing in HH24g mesenchyme, we analysed the KEGG and clustering analyses for pathways/genes predominantly active at late stages of development, and that are maintained in distal mesenchyme transplanted to an earlier environment. This highlighted the TGFβ pathway (*Figure 1e*) and *Bmp2* and *Bmp7* in cluster 1

that encode members of the Bone Morphogenetic Protein (BMP) family, which are signalling ligands of the TGFβ superfamily (*Figure 1f*, *Figure 1—figure supplement 1*, *Supplementary file 3*). Notably, BMP signalling is implicated in terminating outgrowth at the end of the patterning phase by inhibiting FGF signalling by the apical ectodermal ridge (*Pizette and Niswander, 1999*). Readcounts of the RNA-seq data show that the expression of *Bmp2* increases by 5.3-fold between HH24 and HH29 – similar to the difference in expression in HH24g tissue compared to HH24 tissue (6.2-fold - $\log_2$ scale is shown), thus showing that late expression levels of *Bmp2* are maintained in an earlier environment (*Figure 2a*). Similarly, a 1.5-fold increase in *Bmp7* expression occurs between HH24 and HH29, and the difference between expression levels in HH24g tissue compared to HH24 tissue is 2.5-fold (*Figure 2c*). Analyses by RNA in situ hybridization are consistent with these observations

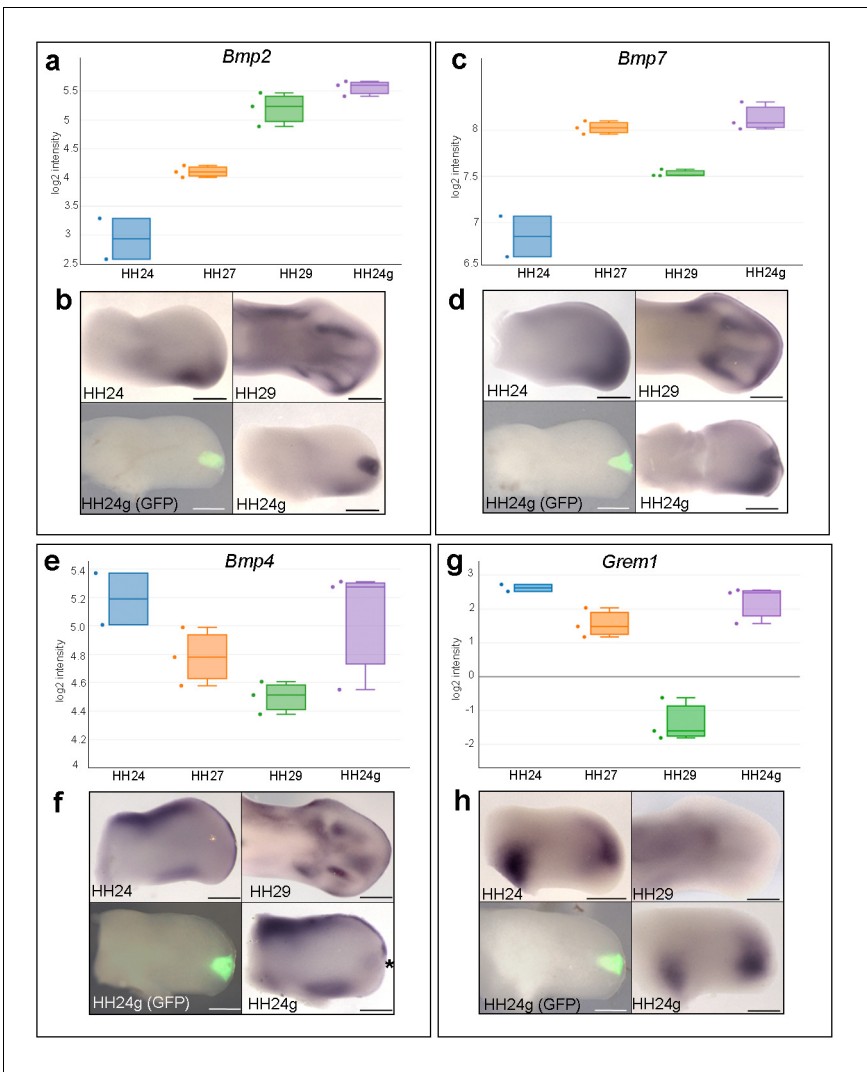

**Figure 2.** *Bmp2/4/7* and *Grem1* expression in HH24g mesenchyme. (**a**) Histogram showing expression levels of *Bmp2* as normalised $\log_2$ values of RNA sequencing read-count intensities. (**b**) In situ hybridization showing *Bmp2* expression - note intense expression in HH24g mesenchyme in area of graft ($n = 3/3$, HH24 is the contralateral bud flipped horizontally). (**c**) Expression levels of *Bmp7* determined by RNA-seq. (**d**) In situ hybridization showing *Bmp7* expression - note intense region of expression in HH24g mesenchyme ($n = 3/3$). (**e**) Expression levels of *Bmp4* determined by RNA-seq. (**f**) In situ hybridizations showing *Bmp4* expression - note enhanced expression in HH24g mesenchyme in area of graft (asterisk - $n = 2/3$). (**g**) Expression levels of *Grem1* determined by RNA-seq. (**h**) In situ hybridizations showing resetting of *Grem1* expression ($n = 7/9$). Scale bars: HH24 buds - 500 μm; HH29 buds - 200 μm.

DOI: https://doi.org/10.7554/eLife.37429.006

(*Figure 2b and d*). We also analysed the expression of *Bmp4* which has a prominent role in limb development (*Selever et al., 2004*; *Bénazet et al., 2009*). The RNA-seq data indicates that *Bmp4* expression levels decrease by two-fold between HH24 and HH29 and that expression levels are approximately equivalent between HH24 and HH24g tissue (*Figure 2e*, note stronger expression in ridge at HH24 compared to *Bmp2/7*). Despite this, slightly enhanced expression could be detected by in situ hybridization in the region of the grafted tissue in HH24g buds (asterisk - *Figure 2f*) in comparison to the equivalent position in contralateral buds. However, cluster 3, which comprises genes that decrease over time and that demonstrate resetting behaviour, contains (*Gremlin1*) *Grem1*, which encodes the BMP antagonist that maintains the apical ectodermal ridge (*Figure 1—figure supplement 3*, *Supplementary file 5*). Between HH24 and HH29, *Grem1* expression levels are reduced by 17.2-fold until they are almost undetectable by in situ hybridization (*Figure 2g and h*). However, *Grem1* expression levels are approximately the same in HH24 and HH24g mesenchyme (*Figure 2g and h*). This result suggests that Grem1 could antagonise BMP2/4/7 signalling in HH24g mesenchyme.

## BMP signalling is maintained in HH24g mesenchyme

To analyse BMP signalling in HH24g mesenchyme, we selected a bona-fide target, *Msx2* (*Pizette et al., 2001*). RNA-seq data shows that *Msx2* has a similar profile to both *Bmp2* and *Bmp7*: a 2.8-fold increase between HH24 and HH29 and a two-fold difference in HH24g tissue compared to HH24 tissue (*Figure 3a and b*). Consistent with this finding, immunofluorescence on sections using an antibody that recognizes phosphorylated SMAD1/5/9 – the signal transducer for BMP2/4/7 – reveals increased levels in later wing buds (*Figure 3c–e*). However, whereas weak expression is detected in HH24 wing buds (*Figure 3c*), strong expression is observed in the distal part of grafts (the area processed for RNA-seq) in contralateral HH24g buds (*Figure 3f*). Control grafts of HH20 distal mesenchyme made to host HH20 wing buds show no obvious differences in pSMAD1/5/9 signal intensity after 24 hr compared to stage-matched HH24 buds (*Figure 3g*–compare to *Figure 3c*). Enhanced pSMAD1/5/9 signal is also detected in the host apical ectodermal ridge above HH24g mesenchyme (asterisk *Figure 3h*), in contrast to adjacent regions (arrow – *Figure 3h*). This result suggests that the level of BMP signalling is maintained in mesenchyme transplanted from older buds to a younger environment, and that it is also enhanced in the host apical ectodermal ridge in response to the older mesenchyme.

BMP signalling is associated with regions of apoptosis in the developing limb, including anterior and posterior necrotic zones, as well as the interdigital mesenchyme (*Zuzarte-Luis and Hurle, 2005*). To determine if apoptosis could reduce proliferation in HH24g mesenchyme, we performed lysotracker staining of apoptotic cells. This reveals no appreciable difference in HH24 control (*Figure 3i*) and HH24g mesenchyme (*Figure 3j*) in whole-mount preparations. Note, the anterior necrotic zone can be observed in both examples (asterisks in *Figure 3i and j*).

To investigate if BMP signalling is responsible for the reduction in proliferation in HH24g mesenchyme (*Saiz-Lopez et al., 2017*), we implanted beads soaked in the BMP antagonist, Noggin, into HH20 wing buds at the time of grafting. After 8 hr, we used flow cytometry to determine the proportion of cells in the different phases of the cell cycle in distal tissue (150 microns from the distal tip). Control PBS- and Noggin-soaked beads implanted into HH20 wing buds for 8 hr do not significantly change distal mesenchyme cell cycle parameters compared to control left wing buds (*Figure 3k*). In HH24g grafts treated with PBS-soaked beads, 69% of distal mesenchyme cells are in G1-phase compared to 60.8% in the equivalent region of the contralateral wing bud (*Figure 3k*). This is consistent with the reduced proliferative potential we previously observed in such grafts (*Figure 1a*, [*Saiz-Lopez et al., 2017*]). However, in HH24g mesenchyme treated with Noggin-soaked beads, the proportion of cells in G1-phase decreases from 69% to 62.4% - a value significantly closer to the control value of 60.8% in the contralateral bud (*Figure 3k*). This result shows that the inhibition of BMP signalling in HH24g mesenchyme can significantly enhance proliferation.

To test if BMP signalling has an instructive effect on cell proliferation in early wing buds, we implanted beads soaked in two concentrations of recombinant BMP2 protein (BMP2$^{high}$ and BMP2$^{low}$ - see Materials and methods) into the distal mesenchyme of HH20 buds, and compared cell cycle profiles to those obtained in contralateral wing buds. After 8 hr, the proportion of cells in G1-phase in control wing buds is 57.8%, but this increases to 60.5% and 66.2% in BMP2$^{low}$ and BMP2$^{high}$ buds, respectively (*Figure 3k*). This finding shows that enhanced BMP signalling inhibits

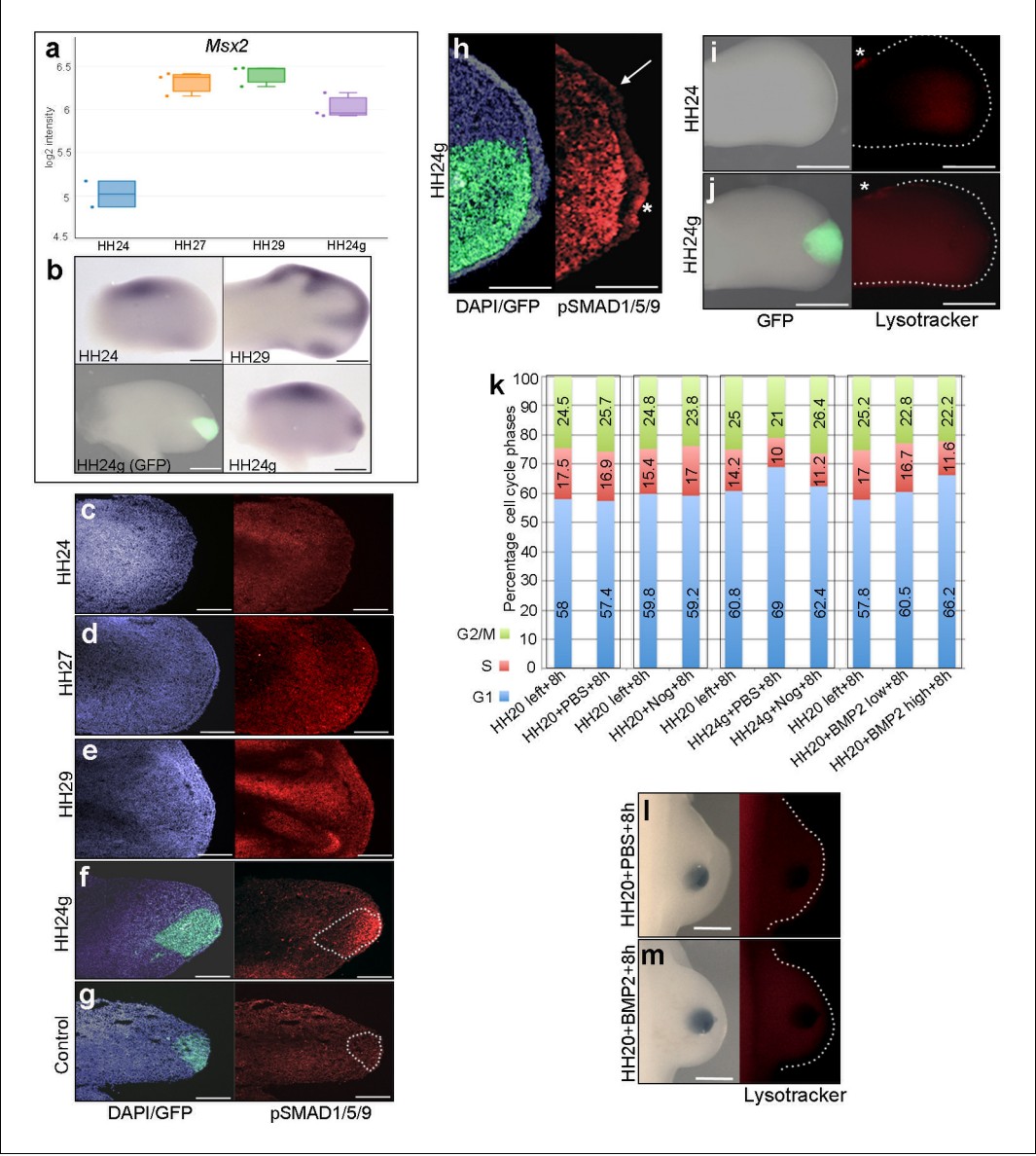

**Figure 3.** BMP signalling in HH24g mesenchyme (a) Expression levels of *Msx2* determined by RNA-seq. (b) In situ hybridization showing *Msx2* expression - note intense expression in HH24g mesenchyme in grafted area (*n* = 3/4). (c–h) Immunostaining of pSMAD1/5/9: signal increases in the distal part of chick wing buds over time (c–e) and is enhanced in HH24g mesenchyme (f) compared to contralateral HH24 buds (*n* = 3/3 c) but not in HH20-HH20 grafts (*n* = 2/2 g); signal is enhanced in apical ectodermal ridge above grafts in HH24g mesenchyme (asterisk-h) but not adjacent regions of ridge (*n* = 2/2 - arrow-h). (i–j) Lysotracker staining of apoptotic cells reveals no difference in HH24 (i) and HH24g (j) mesenchyme (*n* = 5/5) – asterisks indicate anterior necrotic zone. (k) Flow cytometry of wing bud distal mesenchyme: PBS- and Noggin-soaked beads do not significantly affect proportion of cells in G1-phase after 8 hr compared to left wing controls (Pearson's $\chi^2$ test – p=0.1 and p=0.4, respectively); Noggin-soaked beads implanted at HH20 (+Nog) produce a significant decrease in proportion of cells in G1-phase in HH24g grafts compared to PBS-treated controls (+PBS) after 8 hr (Pearson's $\chi^2$ test - p<0.0001); BMP2-soaked beads implanted at HH20 significantly increase G1-phase cells after 8 hr compared to left wing controls (Pearson's $\chi^2$ test - p<0.0001), note boxes indicate separate experiments. (l–m) Lysotracker staining of apoptotic cells reveals no difference in PBS (l) - *n* = 2/2 and BMP2 (m) - *n* = 2/2 treated mesenchyme after 8 hr. Scale bars: HH24 buds - 500 μm, HH29 buds - 200 μm in b; 150 μm in c-g; 75 μm in h; 300 μm in i-j and l-m.

DOI: https://doi.org/10.7554/eLife.37429.007

The following source data is available for figure 3:

**Source data 1.** Flow cytometry graphs for cell cycle analyses

DOI: https://doi.org/10.7554/eLife.37429.008

proliferation of distal mesenchyme cells in the early chick wing bud in a dose-dependent fashion. In addition, lysotracker staining shows that both PBS- (*Figure 3l*) and BMP2[high] -soaked beads (*Figure 3m*) do not cause apoptosis after 8 hr, thus showing that BMP signalling does not inhibit proliferation by compromising cell survival. Taken together, these results suggest that BMP signalling in HH24g mesenchyme maintains an intrinsic cell cycle programme.

## FGF response is reset in HH24g mesenchyme

The maintenance of BMP signalling in HH24g mesenchyme could also reduce proliferation via inhibition of FGFs produced by the apical ectodermal ridge. However, consistent with the observation that *Grem1* expression is maintained in HH24g mesenchyme (*Figure 2g,h*), we have previously shown that *Fgf8* expression is maintained in the apical ectodermal ridge above grafts of distal mesenchyme from much older wing buds - including in tissue that would have developed to stage HH36 if left in situ (or day 10 of incubation – the ridge normally regresses at HH29/30 in the chick wing at day 6–6.5 of incubation - *Saiz-Lopez et al., 2017*). To determine if HH24g distal mesenchyme cells are responsive to apical ectodermal ridge signals, we analysed a well-defined read-out of FGF signalling in the limb, *Dusp6* (*Eblaghie et al., 2003*) - also known as *Mkp3* and *Pyst1* - see also *Figure 4g–j*). RNA-seq read-counts show that *Dusp6* expression decreases by 2.1-fold between HH24 and HH29, but that no significant difference in expression is observed between HH24g tissue and the equivalent region of contralateral control buds, indicating that the level of FGF signalling is approximately the same in the grafted cells as in the host wing bud (*Figure 4a and b*). We also analysed the expression of *Sprouty2* (*Spry2*) that is another well-defined downstream target of FGF signalling (*Chambers and Mason, 2000*). Similar to *Dusp6*, expression of *Spry2* decreases between HH24 and HH29 (1.7-fold), but is approximately equivalent in HH24g compared to control HH24 mesenchyme (*Figure 4c and d*). These data show that *Dusp6* and *Spry2* expression levels decrease over time, consistent with the maturation and regression of the apical ectodermal ridge, and that both are expressed in HH24g mesenchyme at similar level as in surrounding host tissue, thus indicating that grafted cells are responding to FGF signalling. Notably, cluster 3 contains genes encoding other members of the FGF family, including *Fgf13* and *Fgf18*, which are expressed in distal mesenchyme at early stages (*Munoz-Sanjuan et al., 1999*; *Mok et al., 2014*) and that can be reset in a younger environment (*Figure 1—figure supplement 3*, *Supplementary file 5*). RNA-seq read-counts show that *Fgf18* decreases by 90-fold between HH24 and HH29, by which point it is undetectable by in situ hybridization (*Figure 4e and f*), but expression is only 3.4-fold higher in HH24 tissue compared to HH24g tissue. Therefore, taken together, these results show that HH24g mesenchyme is responsive to FGF signalling.

Our data suggest that autocrine BMP signalling is responsible for the intrinsic decline in proliferation in distal mesenchyme of the chick wing bud. However, we also examined if FGF signalling could also have an instructive role. To do this, we implanted beads soaked in recombinant FGF8 protein to the distal mesenchyme of chick wing buds at HH27, the stage at which proliferation rapidly starts to decline (*Figure 1a*) (*Saiz-Lopez et al., 2017*) and determined cell cycle profiles. RNA in situ hybridisation shows that FGF-soaked beads, but not PBS-soaked beads, implanted into HH27 wing buds for 6 hr increase expression of *Dusp6* (*Figure 4g–j*). Flow cytometry reveals that FGF-soaked beads, but not control PBS-soaked beads, significantly increase the proportion of cells in G1-phase compared to control left wing buds (*Figure 4k*– 70.8 vs. 67.8%). Therefore, FGF signalling appears to have a slight inhibitory effect on proliferation, rather than instructively regulating it in a positive manner.

## Discussion

By grafting late mesenchyme cells into an early epithelial environment, we have determined that the progressive loss of e-m signalling is not required for the decline and cessation of chick wing bud outgrowth at the end of the patterning phase. Our data reveals that the duration of mesenchyme cell proliferation is intrinsically determined but requires permissive signals from the apical ectodermal ridge: a distinction not recognized in previous approaches. For instance, surgical removal of the apical ectodermal ridge in the chick wing before HH24 causes apoptosis in the distal mesenchyme, but at later stages decreases proliferation (*Dudley et al., 2002*). In addition, genetic approaches that compromise apical ectodermal ridge signalling in the mouse limb also cause defective outgrowth

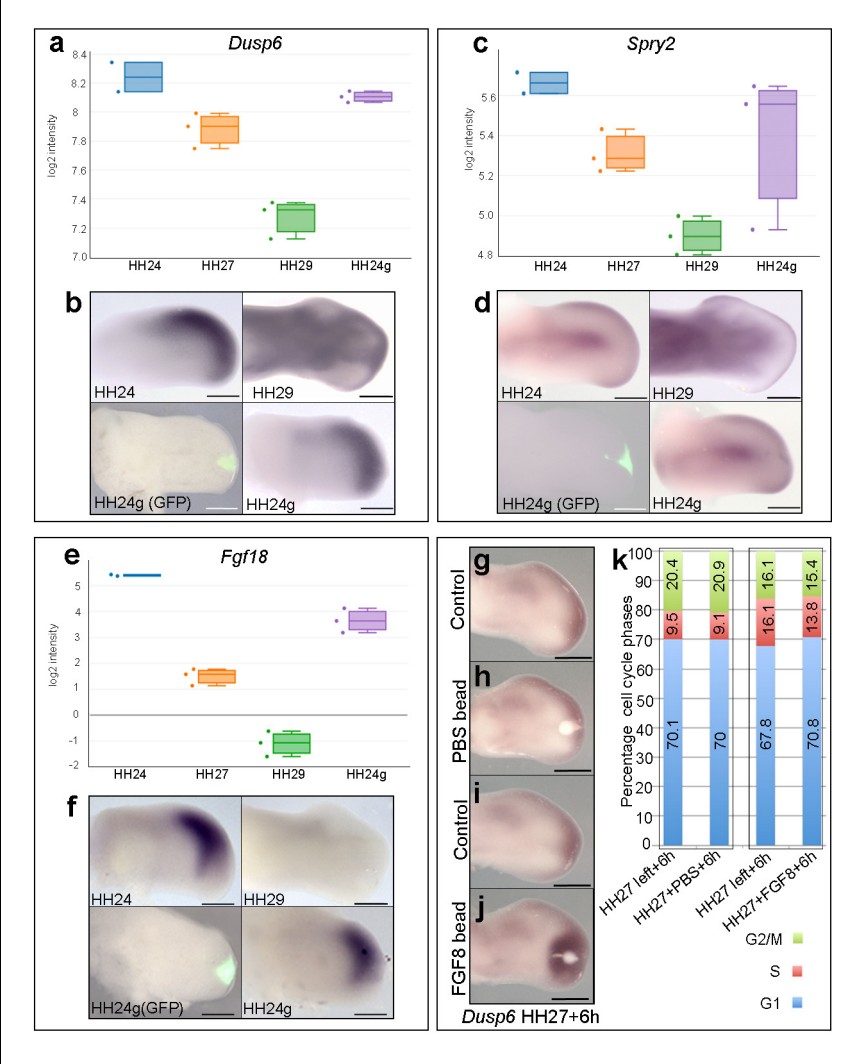

**Figure 4.** FGF signalling in HH24g mesenchyme. (a-e) Resetting of gene expression in HH24g mesenchyme. Expression levels of *Dusp6* (a), *Spry2* (c) and *Fgf18* (e) determined by RNA-seq. (b) In situ hybridization showing equivalent levels of *Dusp6* (n = 3/3, (b), *Spry2* (n = 3/3, (d), and *Fgf18* (n = 3/3, (f) in HH24 and contralateral HH24g mesenchyme. (g–j) FGF-soaked beads (j) n = 3/3), but not PBS-soaked beads (h) n = 3/3), up-regulate expression of *Dusp6* after 6 hr (control left wing buds flipped horizontally – (g) and (i). (k) Flow cytometry of wing bud distal mesenchyme: PBS-soaked beads do not significantly affect proportion of cells in G1-phase after 6 hr compared to left wing controls (Pearson's $\chi^2$ test – p=0.5); FGF-soaked beads implanted at HH27 significantly increase G1-phase cells compared to left wing control after 6 hr (Pearson's $\chi^2$ test - p<0.0001), note boxes indicate separate experiments. Scale bars: HH24 buds, HH27 buds - 300 µm; HH29 buds - 200 µm.
DOI: https://doi.org/10.7554/eLife.37429.009

The following source data is available for figure 4:

**Source data 1.** Flow cytometry graphs for cell cycle analyses
DOI: https://doi.org/10.7554/eLife.37429.010

(*Lu et al., 2008*; *Mariani et al., 2008*; *Yu and Ornitz, 2008*). The fact that mesenchyme expansion was naturally inhibited in these experiments therefore masked the intrinsic control of proliferation in this tissue.

A schematic detailing the relationship between e-m signalling and mesenchyme proliferation during normal wing outgrowth is shown in *Figure 5*. At early stages of wing development (HH20), the Grem1-mediated inhibition of BMP signalling fulfils two main roles: to maintain the apical ectodermal ridge until HH28 as classically-described (*Summerbell, 1974*; *Summerbell et al., 1973*) and to

**Figure 5.** An intrinsic cell cycle timer terminates limb bud outgrowth. HH20-HH24: antagonism of BMP signalling by Grem1 permits rapid mesenchyme proliferation and maintenance of the apical ectodermal ridge (purple) that permissively supports outgrowth. HH24-HH27: intrinsic rise in BMP signalling counters BMP antagonism and instructively decelerates mesenchyme proliferation rate. HH27-HH30: mesenchyme proliferation diminishes as patterning completes at HH28 and then BMP signalling inhibits FGF signalling in apical ectodermal ridge that then regresses.

DOI: https://doi.org/10.7554/eLife.37429.011

permit a rapid rate of proliferation in distal mesenchyme cells as we have shown here. Around HH20, the mechanism of positional specification switches in mesenchyme progenitors from extrinsic signalling (humerus) (*Cooper et al., 2011*; *Roselló-Díez et al., 2011*) to an intrinsic timer (distal to elbow) (*Saiz-Lopez et al., 2015*). This intrinsic mechanism involves a progressive reduction in the proliferation rate as the elements are laid down in a proximal to distal sequence until HH28 (*Saiz-Lopez et al., 2015*) (*Figure 5*). Here, we showed that the deceleration in proliferation rate is caused by an increase in autocrine BMP signalling between HH24 and HH29 that overcomes Grem1-mediated antagonism in distal mesenchyme cells. Once the intrinsic cell cycle timer has run its course at HH29 and distal mesenchyme expansion ceases, high BMP signalling causes the apical ectodermal ridge to regress at HH29/30 (*Figure 5*). Interestingly, recent genetic evidence in the mouse has provided evidence that BMP signalling terminates the extension of the main body axis (*Anderson et al., 2016*).

It was unexpected that all the genes identified in the RNA-seq analysis that are predominantly expressed at early stages could be reset to some degree by the signalling environment (Cluster 3). This is consistent with the observations that early proximo-distal patterning is controlled by extrinsic signals – possibly retinoic acid signalling from the flank of the embryo (*Cooper et al., 2011*; *Roselló-Díez et al., 2011*). Notably, this cluster contains several genes encoding products known to be involved in maintaining a stem-cell-like state including *Plzf* (*Zbtb16*) (*Liu et al., 2016*) *Sall4* (*Yang et al., 2008*) (also *Sall1* and *3*), *Gbx2* (*Tai and Ying, 2013*) and *Lin28b* (*Zhang et al., 2016*). This raises the possibility that a gene regulatory network operates during early limb development to maintain the multipotency of mesenchyme cells and warrants further investigation.

Opposing gradients of BMP ligands and their antagonists specify positional identities and regulate growth in a concentration-dependent manner in many developmental contexts (*Bier and De Robertis, 2015*). Thus, the intrinsic switch from BMP antagonism to BMP signalling in limb development that we describe here provides a mechanism by which cells can be instructed by autocrine BMP signalling in the absence of spatial gradients. It will be important to identify if this mechanism operates in other patterning systems.

# Materials and methods

## Chick husbandry and tissue grafting

Wild-type and GFP-expressing Bovans Brown chicken eggs were incubated and staged according to Hamburger Hamilton (*Hamburger and Hamilton, 1951*) HH19 is embryonic day 3, HH24 - day 4, HH27 - day 5, HH29 – day 6 and HH30, day 7. For tissue grafting, the polarizing region was

discarded and a stripe of 150 μm of distal sub-AER mesenchyme about 500 μm long was dissected from HH27 wing buds. The ectoderm was removed after incubation in 0.25% trypsin at room temperature for 2 min and the mesenchyme was then cut into three cuboidal pieces. These were then placed in slits made using a fine sharpened tungsten needle between the apical ectodermal ridge and underlying mesenchyme in the mid-distal region of HH20 wing buds.

## RNA sequencing analyses and clustering

Grafts were made as described above and after 24 hr the 150 μm distal part of the graft was carefully dissected under a UV scope. Equivalent tissue was dissected from the contralateral HH24 wing bud and also 150 μm of tissue was dissected from three positions at the distal tips of HH24, HH27 and HH29 wing buds. Three replicate experiments were performed from each condition and the tissue was pooled before the RNA was extracted using Trizol reagent (Gibco), and sequenced on a HiSeq 2000 (Paired end readings of 50 bp - Instrument: ST300). Sequencing data was mapped using Hisat v2.0.3 (bam file generation). Salmon output was used to quantify the data (number of reads) (*Patro et al., 2017*). The raw data has been deposited in array express (https://www.ebi.ac.uk/arrayexpress/experiments/E-MTAB-6437/). Twelve samples (three for each of the four conditions HH24, HH24g, HH27 and HH29) were checked for quality issues. This was done by manually inspecting the density plot, boxplots, PCA plots, correlation heatmap and distance plot, as well as using several automatic outlier tests, namely distance, Kolmogorov-Smirnov, correlation and Hoeffding's D. Based on this, one of the HH24 samples was excluded from the analysis.

The count data for the samples were normalised using trimmed mean of $m$-value normalisation and transformed with Voom, resulting in $\log_2$-counts per million with associated precision weights (*Robinson and Oshlack, 2010*; *Law et al., 2014*). A heat-map was made showing the correlation (Pearson) of the normalised data collapsed to the mean expression per group. A statistical analysis using an adjusted p-value<0.0005 and a fold-change >2 identified 151 genes as differentially expressed in the two contrasts evaluated. Gene clusters were identified from the set of 151 unique genes that were differentially expressed in at least one of the statistical comparisons. The evaluation considered between two and 35 clusters using hierarchical, $k$-means, and PAM clustering methods based on the internal, stability and biological metrics provided from the clValid R package. The majority of internal validation and stability metrics indicated that either the lowest possible number or conversely the highest number evaluated were preferable. The metrics giving more nuanced information in the intermediate range were the Silhouette measure, and the Biological Homogeneity Index (BHI). Based on manual inspection, it was decided to use hierarchical clustering with $k = 3$ gene clusters, which showed favourable properties for both these measures. For KEGG pathway analyses, significant genes with adjusted p-value<0.05 and fold-change >= 2 from each comparison were analysed for enrichment of KEGG pathway membership using a hypergeometric test. Enrichment (p<0.05) was assessed for up and down-regulated genes separately.

## Apoptosis analyses

Chick wing buds were dissected in PBS and transferred to Lysotracker (Life Technologies, L-7528) PBS solution (1:1000) in the dark. Wing buds were incubated for 1 hr at 37°C, washed in PBS, and fixed overnight in 4% PFA at 4°C. Wing buds were then washed in PBS and progressively dehydrated in a methanol series.

## Whole mount in situ hybridisation

Embryos were fixed in 4% PFA overnight at 4°C, dehydrated in methanol overnight at −20°C, rehydrated through a methanol/PBS series, washed in PBS, then treated with proteinase K for 20 min (10 μg/ml$^{-1}$), washed in PBS, fixed for 30 mins in 4% PFA at room temperature and then prehybridised at 69°C for 2 hr (50% formamide/50% 2x SSC). 1 μg of antisense DIG-labelled mRNA probes were added in 1 ml of hybridisation buffer (50% formamide/50% 2x SSC) at 69°C overnight. Embryos were washed twice in hybridisation buffer, twice in 50:50 hybridisation buffer and MAB buffer, and then twice in MAB buffer, before being transferred to blocking buffer (2% blocking reagent 20% lamb serum in MAB buffer) for 2 hr at room temperature. Embryos were transferred to blocking buffer containing anti-digoxigenin antibody (1:2000) at 4°C overnight, then washed in MAB buffer

overnight before being transferred to NTM buffer containing NBT/BCIP and mRNA distribution visualised using a LeicaMZ16F microscope.

## Immunohistochemistry

Embryos were fixed in 4% PFA for 2 hr on ice, washed in PBS and transferred to 30% sucrose overnight at 4°C. Embryos were frozen in OCT mounting medium, fixed to a chuck and cryosectioned immediately. Sections were dried for 2 hr and washed in PBS containing 0.1% triton for 10 min at room temperature. Sections were blocked in 0.1% Triton, 1 – 2% hings/serum for 1 – 1.5 hr at room temperature. Primary antibody solution was added (1:500 rabbit anti-PSMAD1/5/9 Cell Signalling Technology – D5B10) in PBS/0.1% triton/1 – 2% hings, before coverslips were added and slides placed in a humidified chamber at 4°C for 72 hr. Slides were then washed for $3 \times 5$ min in PBS at room temperature. Secondary antibody (1:500 anti-rabbit Alexa Fluor 594 Cell Signalling Technology – 8889S) was added in PBS 0.1% triton/1 – 2% hings for 1 hr at room temperature. Slides were washed in PBS for $3 \times 5$ min and mounted in Vectashield/DAPI medium. Slides were left at 4°C and imaged the following day.

## Flow cytometry

Distal 150 μm blocks of mesenchyme tissue were dissected in ice cold PBS under a LeicaMZ16F microscope using a fine surgical knife pooled from replicate experiments (12), and digested into single-cell suspensions with trypsin (0.5%, Gibco) for 30 min at room temperature. Cells were briefly washed twice in PBS, fixed in 70% ethanol overnight, washed in PBS and re-suspended in PBS containing 0.1% Triton X-100, 50 μg/ml$^{-1}$ of propidium iodide and 50 μg/ml$^{-1}$ of RNase A (Sigma). Dissociated cells were left at room temperature for 20 min, cell aggregates were removed by filtration and single cells analysed for DNA content with a FACSCalibur flow cytometer and FlowJo software (Tree star Inc). Based on ploidy values cells were assigned in G1-, S-, or G2/M-phases and this was expressed as a percentage of the total cell number (>7500 cells in each case). Statistical significance of numbers of cells between pools of dissected wing bud tissue (10–12 in each pool) was determined by Pearson's $\chi^2$ tests to obtain two-tailed p-values (significantly different being a p-value of less than 0.05 - see (*Chinnaiya et al., 2014*) - statistical comparisons of cell cycle parameters between the left and right wing buds embryos showed that differences are less than 1%).

## Bead implantation

Control affigel blue beads (Biorad) were soaked in PBS/4 mM HCl. Affigel beads were soaked in Human Noggin (0.025 μg μl$^{-1}$ - R and D) or human BMP2 protein (0.025 μg μl$^{-1}$ and 0.05 μg μl$^{-1}$ - R and D) dissolved in PBS/4 mM HCl. Control heparin beads (Biorad) were soaked in PBS. Heparin beads were soaked in human FGF8 (0.5 μg μl$^{-1}$ - R and D dissolved in PBS. Beads were soaked for 2 hr and implanted into the distal mesenchyme using a sharp needle.

## Acknowledgements

We thank the Wellcome Trust for funding, Cheryll Tickle for critical reading, Adrian Sherman/Helen Sang for GFP-expressing chicken embryos, the University of Sheffield flow cytometry core facility and Max Bylesjo and Paul McAdam at FIOS Genomics for bioinformatics.

## Additional information

### Funding

| Funder | Grant reference number | Author |
| --- | --- | --- |
| Wellcome Trust | 202756/Z/16/Z | Joseph Pickering<br>Constance A Rich<br>Holly Stainton<br>Kavitha Chinnaiya<br>Matthew Towers |

| Spanish Ministerio de Economia | BFU2017-88265- P | Cristina Aceituno<br>Patricia Saiz-Lopez<br>Marian A Ros |
|---|---|---|

The funders had no role in study design, data collection and interpretation, or the decision to submit the work for publication.

## Author contributions
Joseph Pickering, Formal analysis, Validation, Investigation, Visualization, Methodology, Writing—original draft; Constance A Rich, Formal analysis, Validation, Investigation; Holly Stainton, Cristina Aceituno, Kavitha Chinnaiya, Formal analysis, Investigation; Patricia Saiz-Lopez, Conceptualization, Investigation, Methodology; Marian A Ros, Formal analysis, Supervision, Validation, Investigation, Methodology, Project administration, Writing—review and editing; Matthew Towers, Conceptualization, Formal analysis, Supervision, Funding acquisition, Validation, Methodology, Writing—original draft, Project administration, Writing—review and editing

## Author ORCIDs
Joseph Pickering (iD) http://orcid.org/0000-0002-5892-5159
Kavitha Chinnaiya (iD) http://orcid.org/0000-0002-3375-420X
Patricia Saiz-Lopez (iD) http://orcid.org/0000-0001-7106-5192
Marian A Ros (iD) http://orcid.org/0000-0002-1224-7671
Matthew Towers (iD) http://orcid.org/0000-0003-2189-4536

## Decision letter and Author response
Decision letter https://doi.org/10.7554/eLife.37429.021
Author response https://doi.org/10.7554/eLife.37429.022

# Additional files

## Supplementary files
• Supplementary file 1. RNA sequencing of HH24g vs. HH24 distal mesenchyme 55 genes are differentially expressed between HH24g and HH24 distal mesenchyme. 53 genes are increased in expression in HH24g distal mesenchyme and two genes are decreased in expression.
DOI: https://doi.org/10.7554/eLife.37429.012

• Supplementary file 2. RNA sequencing of HH24g vs. HH29 distal mesenchyme 99 genes are differentially expressed between HH24g and HH29 distal mesenchyme. 55 genes are increased in expression in HH24g distal mesenchyme and 44 genes are decreased in expression.
DOI: https://doi.org/10.7554/eLife.37429.013

• Supplementary file 3. Cluster 1 Clustering of RNA-seq data across HH24g/HH24 and HH24g/HH29 pairwise contrasts. Cluster 1: genes that increase between HH24 and HH29 (black line) and are maintained in HH24g distal mesenchyme.
DOI: https://doi.org/10.7554/eLife.37429.014

• Supplementary file 4. Cluster 2 Clustering of RNA-seq data across HH24g/HH24 and HH24g/HH29 pairwise contrasts. Cluster 2: genes that increase between HH24 and HH29 (black line) and reset in HH24g distal mesenchyme.
DOI: https://doi.org/10.7554/eLife.37429.015

• Supplementary file 5. Cluster 3 Clustering of RNA-seq data across HH24g/HH24 and HH24g/HH29 pairwise contrasts. Cluster 3: genes that decrease between HH24 and HH29 (black line) and reset in HH24g distal mesenchyme.
DOI: https://doi.org/10.7554/eLife.37429.016

• Transparent reporting form
DOI: https://doi.org/10.7554/eLife.37429.017

## Data availability

RNA sequencing data has been deposited (https://www.ebi.ac.uk/arrayexpress/experiments/E-MTAB-6437/).

The following dataset was generated:

| Author(s) | Year | Dataset title | Dataset URL | Database, license, and accessibility information |
|-----------|------|---------------|-------------|--------------------------------------------------|
| Joseph Pickering, Matthew Towers | 2018 | RNA sequencing of chicken limb distal mesenchyme cells to compare gene expression over time and in heterochronic grafted cells | https://www.ebi.ac.uk/arrayexpress/experiments/E-MTAB-6437/ | Publicly available at the Electron Microscopy Data Bank (accession no: E-MTAB-6437) |

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
