## [Decision Letter]

[Editors’ note: the authors were asked to provide a plan for revisions before the editors issued a final decision. What follows is the editors’ letter requesting such plan.]

Thank you for sending your article entitled "An intrinsic cell cycle timer terminates limb bud outgrowth" for peer review at *eLife*. Your article is being reviewed by Marianne Bronner as the Reviewing and Senior Editor and three reviewers.

Given the list of essential revisions, including new experiments, the editors and reviewers invite you to respond with an action plan and timetable for the completion of the additional work. We plan to share your responses with the reviewers and then issue a binding recommendation.

Summary:

In this paper, the authors attempt to show that the breakdown of reciprocal extrinsic signaling between the mesenchyme and the overlying epithelium is not required for the termination of proliferative outgrowth of limb. For this purpose, they performed RNA-seq of distal cells from HH24, HH27, HH29 wing buds as well as HH24g (grafts of old mesenchyme cells (HH27) to early HH20 wing buds, which is left until HH24). RNA-seq analyses revealed that, in HH24g, Bmp2/4/7 expression is maintained, whereas FGF and Grem1 expression is reset. Based on these results, authors draw conclusions that the mesenchyme cells terminate growth by an intrinsic cell cycle timer in the presence of e-m signalling. While the results are interesting, many changes are required to bring the manuscript to publishable form. Most critical is to perform further functional experiments to test the role of genes identified by RNA-seq to validate the model. It would also be critical to provide evidence that the permissive signals from the AER is required for the duration of mesenchyme cell proliferation. The major issues raised by the reviewers are listed below. If you think you can make these changes in 2-3 months, we would be willing to entertain a revised version of the manuscript. To this end, it would be very helpful if you could provide us with an outline of how you would address the reviewers' criticisms.

Essential revisions:

1) Three criteria for demonstrating a class of factors (e.g., BMPs) is responsible for an event (e.g., loss of proliferation in HH24g cells). (1) Is it present at the right place and time? Figure 2 addresses this. (2) Can it perform this function? Figure 3 K-L addresses this. (3) Removal of the factor should block the event. An argument can be made that the last criterion is most important. There is no experiment in this study that meets this criterion. Therefore, additional functional data are essential; otherwise this study remains much too inconclusive. Perhaps a Noggin bead, an siRNA-inhibition of Bmpr1A experiment, or a chemical inhibition of BMP signaling such as LDN-193189 can be used to test this idea. I think this is the biggest issue and needs to be addressed before publication.

2) The authors never use the term "autocrine" in regard to BMP signaling in their model. But is it not the case? Please either use the term or explain why it is not appropriate.

3) As the authors point out, it is important to exclude that a loss of AER-FGF is affecting cell cycle in the graft- especially because they find increased pSmad staining in the AER over their grafts. Previous work (Development 144, 479) demonstrates that AER-FGF8 expression persists in such grafts. But it is possible that within the mesenchyme an intrinsic timer regulates its competence to respond to the AER-FGF signal. Is *Dusp6* enough to demonstrate these cells receive a normal FGF signal? Different targets likely respond to different signaling thresholds. The authors should also examine Spry2 and Spry4 in their grafts.

4) Given the well-known effect of limb-bud BMP signalling on cell death, the authors should report what cell death occurred in their bead/flow cytometry experiment (Figure 3K-L).

5) The authors need to expound some about why they propose BMP signaling is instructive and FGF signals are permissive. The "young" AER-FGF signal over the graft has no effect on the cell cycle – hence it is permissive. The authors need to provide further evidence that the permissive signals from the AER is required for the duration of mesenchyme cell proliferation. Why is the BMP signal instructive – a dose response experiment, where more BMP has a greater effect, would help clarify the use of this term.

6) In this work, authors showed that, in HH24g, (a) Bmp2/4/7 expression is maintained (as late stage), (b) FGF and Grem1 expression is reset (as early stage) and (c) implanted (not endogenous) BMP represses cell proliferation in limb buds. Based on these data, authors concluded "an intrinsic switch from BMP antagonism to BMP signalling controls the cell cycle program in distal mesenchyme during limb bud outgrowth". There is a wild leap of logic in this conclusion from limited data presented here.

7) Regarding the Bmp implantation experiment, concentration of Bmp or method was not included. Does this experiment reflect the function of endogenous Bmp on cell proliferation in limb buds?

8) The term "intrinsic cell cycle timer" needs more explanation. Did authors mean that cells memorize the positional information as an epigenetic memory, gene expression, protein accumulation, or cell adhesion? If they consider all these possibilities, other conclusions should also be considered.

9) There are no statistical or scientific reasons why authors selected Bmp, FGF, and Grem1 for further analyses. Please justify why you are looking at BMPs, FGFs, Gremlin. If these are the "usual suspects", one could disregard the RNA seq data. If other enriched genes, such as cyclinD1, BAMBI, SPON1, HOXC13, Sall and EphA4, were taken into consideration, their conclusions might have been changed.

From Cluster 1, authors selected Bmp2 and Bmp7. In the text, it was said that "This cluster notably includes genes encoding members of the Bone Morphogenetic Protein (BMP) family-Bmp2 and Bmp7. Genes encoding other known signaling molecules are not present in cluster1". I do not understand why authors did not perform GO, or similar analyses to select candidates. As authors are interested in the mechanisms of intrinsic cell cycle timer, genes related to proliferation or epigenetic modifications should be also considered. Following genes seem to be highly relevant.

CCND1 (cyclin1)

SPON1(spondin1, a heparin-binding extracellular matrix glycoprotein, which can control bone metabolism, via possible disruption of latent TGF-β activation.)

ARID5B (Transcription coactivator that make complex with PHF2, which mediates demethylation of H3K9me2)

CREB5 (Binds DNA as a homodimer or as a heterodimer with JUN or ATF2/CREBP1)

From Cluster 2, authors selected Fgfr2, but again they did not explain the statistical reasons. Cluster 2 contains the following interesting genes, but not selected for further analyses.

BAMBI (BMP and Activin Membrane Bound Inhibitor)

HoxC13 (Controls transcription of genes, involved cell proliferation, such as CCND1)

FBLN1 (a calcium binding ECM and plasma glycoprotein. Required for normal limb morphogenesis)

RDH10 (retinol dehydrogenase 10. Required for normal limb morphogenesis)

LGR5 (GPCR class A receptor. A member of the Wnt signaling pathway. R-spondin proteins are their ligands)

From cluster 3, they selected FGF and Grem1, again without any statistical reasons. Cluster 3 contains other genes known to be involved in setting positional values of limbs or limb morphogenesis, such as Sall1, 3, 4, EphA4 (receptor for ephrin-A2, which regulates position-specific cell affinity in the chicken limb bud), Dlx2 (TF, Required for limb development). Sall1 and Sall3 contribute to the autopod development, by competing with Hoxa13-d13 for a target sequence in the EphA4 upstream region. Such an antagonistic interaction between Hox and Sall in the limb suggested to be contributed to the fine-tuning of local Hox activity leading to proper morphogenesis of cartilage element (i.e. controlling the positional value of the limb along the proximo-disal axis).

[Editors' note: further revisions were requested prior to acceptance, as described below.]

Thank you for resubmitting your work entitled "An intrinsic cell cycle timer terminates limb bud outgrowth" for further consideration at *eLife*. Your revised article has been favorably evaluated by Marianne Bronner (Senior Editor) and three reviewers.

The manuscript has been improved but there are some issues remaining that need to be addressed before acceptance, as outlined below:

1) The paper ends with "It will be important to identify if this mechanismoperates in other patterning systems." Considering "this mechanism" as BMP signaling that terminates growth, I strongly suggest the authors consider citing Anderson et al., 2016. This paper provides genetic evidence that BMP signals terminate axis extension (Figure 10 is a summary model). The paragraph at the top of page 20 discusses this idea.

2) KEGG analyses

i) It's not clear if KEGG is suitable for identifying pathways in *Gallus gallus*. According to Supplementary file 2, KEGG recognizes pathways for only 24 genes out of 99 genes (fail to recognize SALL3, GREM1, DLX2, EPHA4 etc.). This fact is not even mentioned in the text. ii) Supplementary file names are still not organized. For example, Supplementary file 2 is named "5159_1_supp_876279_pc44ty.xls", and thus I was not able to tell which file is for HH24g-HH24 pair or HH24g-HH29 pair. iii) Description of KEGG analyses is incomplete in Supplementary files. For example, in Supplementary file 2, KEGG Pathways of NPY2R is described as "Neuroactive ligand-receptor in…". iv) Description of KEGG pathways in Supplementary fileSupplementary file 1, Supplementary file 2 and Figure 1E is not consistent. For example, in Supplementary file 2, KEGG Pathways of KCNQ1 is described as "Adrenergic signaling in cardio…", but in Figure 1E, it was "Adrenergic". This should be explained at least in Figure legend. v) In Figure 1E, HH24g vs. HH29 is on the left side in Up-regulated pathways, but on the right side in Down-regulated pathways. It is confusing. They should be on the same side.

3) BMP and Noggin beads implantation.i) In Figure 3K, control for HH24g+Nog+8h is not appropriate. Controls should be limb buds implanted with PBS/4mM HCl (i.e. vehicle)-soaked beads (i.e. HH24g-PBS/4mM HCl+8h). ii) In Figure 3K, controls for HH20+BMP2low/higH^+^8h are not appropriate. Controls should be limb buds implanted with PBS/4mM HCl -soaked beads (i.e. HH20-PBS/4mM HCl+8h). iii) In Figure 3L, control is not appropriate. Again, control should be limb buds implanted with PBS/4mM HCl^-^soaked beads.

4) FGF beads implantation. i) Vehicle for FGF beads implantation is not mentioned in MM. ii) In Figure 4G, control should be limb buds implanted with vehicle-soaked beads. iii) In Figure 4I, again, control should be HH27+vehicle+6h.

---

## [Author Response]

[Editors’ notes: the authors’ response after being formally invited to submit a revised submission follows.]

Summary:In this paper, the authors attempt to show that the breakdown of reciprocal extrinsic signaling between the mesenchyme and the overlying epithelium is not required for the termination of proliferative outgrowth of limb. For this purpose, they performed RNA-seq of distal cells from HH24, HH27, HH29 wing buds as well as HH24g (grafts of old mesenchyme cells (HH27) to early HH20 wing buds, which is left until HH24). RNA-seq analyses revealed that, in HH24g, Bmp2/4/7 expression is maintained, whereas FGF and Grem1 expression is reset. Based on these results, authors draw conclusions that the mesenchyme cells terminate growth by an intrinsic cell cycle timer in the presence of e-m signalling. While the results are interesting, many changes are required to bring the manuscript to publishable form. Most critical is to perform further functional experiments to test the role of genes identified by RNA-seq to validate the model. It would also be critical to provide evidence that the permissive signals from the AER is required for the duration of mesenchyme cell proliferation. The major issues raised by the reviewers are listed below. If you think you can make these changes in 2-3 months, we would be willing to entertain a revised version of the manuscript. To this end, it would be very helpful if you could provide us with an outline of how you would address the reviewers' criticisms.Essential revisions:1) Three criteria for demonstrating a class of factors (e.g., BMPs) is responsible for an event (e.g., loss of proliferation in HH24g cells). (1) Is it present at the right place and time? Figure 2 addresses this. (2) Can it perform this function? Figure 3 K-L addresses this. (3) Removal of the factor should block the event. An argument can be made that the last criterion is most important. There is no experiment in this study that meets this criterion. Therefore, additional functional data are essential; otherwise this study remains much too inconclusive. Perhaps a Noggin bead, an siRNA-inhibition of Bmpr1A experiment, or a chemical inhibition of BMP signaling such as LDN-193189 can be used to test this idea. I think this is the biggest issue and needs to be addressed before publication.

We will graft distal mesenchyme cells from HH27 wing buds to HH20 wing buds and implant Noggin-soaked beads into the host to investigate if this enhances cell cycle progression in the graft. Cell cycle profiles will be compared to the contralateral wing bud, to wings with grafts without Noggin, and to wings treated with Noggin alone.

Revision: We have done this experiment (Figure 3K) and found that after 8h, Noggin-soaked beads significantly reverse the proliferative defect in HH24g mesenchyme. This provides strong evidence that BMP signalling is responsible for loss of proliferation in this tissue.

2) The authors never use the term "autocrine" in regard to BMP signaling in their model. But is it not the case? Please either use the term or explain why it is not appropriate.

The autocrine effect of BMP signalling will be considered and discussed.

Revision: We realise now that we used autocrine synonymously with intrinsic. We have now used autocrine in the text where necessary.

3) As the authors point out, it is important to exclude that a loss of AER-FGF is affecting cell cycle in the graft- especially because they find increased pSmad staining in the AER over their grafts. Previous work (Development 144, 479) demonstrates that AER-FGF8 expression persists in such grafts. But it is possible that within the mesenchyme an intrinsic timer regulates its competence to respond to the AER-FGF signal. Is Dusp6 enough to demonstrate these cells receive a normal FGF signal? Different targets likely respond to different signaling thresholds. The authors should also examine Spry2 and Spry4 in their grafts.

*Dusp6* is a good read-out of FGF signalling since its expression is regulated by active FGF signalling during development and induced in response to an FGF-soaked bead in the distal mesenchyme (we will show this data). We have looked at the RNA sequencing data and *Spry1, 2* and *4* behave like *Dusp6*. We will analyse the expression of *Spry* genes in HH24g wings.

Revision: We have now shown that *Dusp6* is induced in response to FGF8-soaked beads (Figures 4G and H). We decided to analyse the expression of *Spry2* in HH24g wing buds and we now show that this behaves like *Dusp6* (Figures 4C and D).

4) Given the well-known effect of limb-bud BMP signalling on cell death, the authors should report what cell death occurred in their bead/flow cytometry experiment (Figure 3K-L).

We will perform apoptosis analyses on HH21 wing buds treated with BMP2-soaked beads for 6 hours.

Revision: We have now done this experiment and shown that BMP2 beads do not induce cell death (Figures 3L and M). The only thing we changed was to implant beads at HH20 for 8h to maintain consistency with the Noggin-bead experiments.

5) The authors need to expound some about why they propose BMP signaling is instructive and FGF signals are permissive. The "young" AER-FGF signal over the graft has no effect on the cell cycle – hence it is permissive. The authors need to provide further evidence that the permissive signals from the AER is required for the duration of mesenchyme cell proliferation. Why is the BMP signal instructive – a dose response experiment, where more BMP has a greater effect, would help clarify the use of this term.

We will perform an experiment to test if BMP-soaked beads have a dose-dependent effect on cell cycle progression. To further test the permissive role of FGF signalling we will maintain FGF signalling using beads at late stages of chick wing development when the apical ridge begins to regress and see if this can enhance cell cycle progression in an instructive manner. The results of experiments 1 and 3 will also contribute to distinguishing between the permissive and instructive effects of FGF signalling.

Revision: We have now shown that BMP2 beads implanted into HH20 wing buds for 8h have a dose-dependent effect on cell cycle progression suggesting an instructive role (Figure 3K). We have also implanted beads soaked in FGF just as proliferation starts to decline rapidly (HH27) and found that this does not increase cell cycle progression (Figure 4I). This suggests that FGFs play a permissive role in proliferation.

6) In this work, authors showed that, in HH24g, (a) Bmp2/4/7 expression is maintained (as late stage), (b) FGF and Grem1 expression is reset (as early stage) and (c) implanted (not endogenous) BMP represses cell proliferation in limb buds. Based on these data, authors concluded "an intrinsic switch from BMP antagonism to BMP signalling controls the cell cycle program in distal mesenchyme during limb bud outgrowth". There is a wild leap of logic in this conclusion from limited data presented here.

Depending on the results of the above experiments we might have to consider revising these statements, which we agree are too strong at present. We will also further discuss our 2015 paper (Saiz-Lopez et al., 2015) in which we showed that cell cycle progression is intrinsically timed in the distal mesenchyme.

Revision: In our previous paper we showed that the cell cycle is intrinsically regulated although it was unclear how it is terminated (Introduction, Results section). Having now shown that BMP signalling/antagonism controls the cell cycle programme over time we are confident with our speculation. However, we will still say ‘provide evidence’ as to not be too unnecessarily definitive.

7) Regarding the Bmp implantation experiment, concentration of Bmp or method was not included. Does this experiment reflect the function of endogenous Bmp on cell proliferation in limb buds?

We will include information about BMP concentration. The result of the Noggin-bead experiment (point 1) should allow us to speculate more on the role of endogenous BMP signalling.

Revision: We have included details of concentration now and we feel that experiment 1 provides strong evidence for the endogenous role of BMP signalling in the limb.

8) The term "intrinsic cell cycle timer" needs more explanation. Did authors mean that cells memorize the positional information as an epigenetic memory, gene expression, protein accumulation, or cell adhesion? If they consider all these possibilities, other conclusions should also be considered.

We use the term “intrinsic timer” to explain that cell cycle parameters are controlled intrinsically in the mesenchyme, but still require permissive signals from the apical ridge. This was a major part of our 2015 paper and we will explain this better in our revision. We do not yet know what the nature of the clock is.

Revision: At this stage we do not feel we have enough data to speculate upon the potential mechanism of the timer.

9) There are no statistical or scientific reasons why authors selected Bmp, FGF, and Grem1 for further analyses. Please justify why you are looking at BMPs, FGFs, Gremlin. If these are the "usual suspects", one could disregard the RNA seq data. If other enriched genes, such as cyclinD1, BAMBI, SPON1, HOXC13, Sall and EphA4, were taken into consideration, their conclusions might have been changed.From Cluster 1, authors selected Bmp2 and Bmp7. In the text, it was said that "This cluster notably includes genes encoding members of the Bone Morphogenetic Protein (BMP) family-Bmp2 and Bmp7. Genes encoding other known signaling molecules are not present in cluster1". I do not understand why authors did not perform GO, or similar analyses to select candidates. As authors are interested in the mechanisms of intrinsic cell cycle timer, genes related to proliferation or epigenetic modifications should be also considered. Following genes seem to be highly relevant.CCND1 (cyclin1)SPON1(spondin1, a heparin-binding extracellular matrix glycoprotein, which can control bone metabolism, via possible disruption of latent TGF-β activation.)ARID5B (Transcription coactivator that make complex with PHF2, which mediates demethylation of H3K9me2)CREB5 (Binds DNA as a homodimer or as a heterodimer with JUN or ATF2/CREBP1)From Cluster 2, authors selected Fgfr2, but again they did not explain the statistical reasons. Cluster 2 contains the following interesting genes, but not selected for further analyses.BAMBI (BMP and Activin Membrane Bound Inhibitor)HoxC13 (Controls transcription of genes, involved cell proliferation, such as CCND1)FBLN1 (a calcium binding ECM and plasma glycoprotein. Required for normal limb morphogenesis)RDH10 (retinol dehydrogenase 10. Required for normal limb morphogenesis)LGR5 (GPCR class A receptor. A member of the Wnt signaling pathway. R-spondin proteins are their ligands)From cluster 3, they selected FGF and Grem1, again without any statistical reasons. Cluster 3 contains other genes known to be involved in setting positional values of limbs or limb morphogenesis, such as Sall1, 3, 4, EphA4 (receptor for ephrin-A2, which regulates position-specific cell affinity in the chicken limb bud), Dlx2 (TF, Required for limb development). Sall1 and Sall3 contribute to the autopod development, by competing with Hoxa13-d13 for a target sequence in the EphA4 upstream region. Such an antagonistic interaction between Hox and Sall in the limb suggested to be contributed to the fine-tuning of local Hox activity leading to proper morphogenesis of cartilage element (i.e. controlling the positional value of the limb along the proximo-disal axis).

We performed the RNA sequencing experiments to gain some general insights into the reversibility and stability of gene expression. When we realised that *Grem1* was in a cluster of ‘reset’ genes, our focus turned to the mechanism of limb bud growth termination, as this was an unexpected finding. We are very much interested in looking at further classes of genes – particularly Hox genes and Ephrins – that could help us understand the programme that specifies autopod development. In our previous paper in 2017, we showed that autopod fate is maintained in HH24g cells.

We have performed both GO and KEGG analyses and we should have shown this data. There are four up-regulated pathways in HH24g wings compared to HH24 wings. These are: ECM, focal adhesion, adrenergic signalling, and TGF-β signalling. This is why we focussed on BMPs. Surprisingly the RNA sequencing has not provided any indication of cell cycle behaviour – even during the interval that cell proliferation declines in normal development (HH24 to HH29). We think this is because the activities of key cell cycle regulators, such as cyclins, are generally controlled post-translationally.

Revision: We included the KEGG analyses (Figure 1E) as this provided a clearer and simplified representation of altered pathway behaviour than the GO analyses.

[Editors' note: further revisions were requested prior to acceptance, as described below.]

The manuscript has been improved but there are some issues remaining that need to be addressed before acceptance, as outlined below:1) The paper ends with "It will be important to identify if this mechanismoperates in other patterning systems." Considering "this mechanism" as BMP signaling that terminates growth, I strongly suggest the authors consider citing Anderson et al., 2016. This paper provides genetic evidence that BMP signals terminate axis extension (Figure 10 is a summary model). The paragraph at the top of page 20 discusses this idea.2) KEGG analysesi) It's not clear if KEGG is suitable for identifying pathways in Gallus gallus. According to Supplementary file 2, KEGG recognizes pathways for only 24 genes out of 99 genes (fail to recognize SALL3, GREM1, DLX2, EPHA4 etc.). This fact is not even mentioned in the text.

In the chicken genome ~20% of genes belong to verified KEGG pathways. This is slightly less than some model organisms (human, mouse and rat have ~30% annotated), but on par with others (e.g. zebrafish ~20%). Since our KEGG analysis successfully and independently identified the TGFβ and FGF signalling pathways that are the focus of this work, we would prefer to keep the analysis. We also manually inspected the RNA-seq gene lists for other possible candidates involved in maintaining cell cycle progression in HH24g wing buds and we did not find stronger ones than *Bmp2* and *Bmp7*. However, we have mentioned in the Results section that the KEGG analysis is limited.

ii) Supplementary file names are still not organized. For example, Supplementary file 2 is named "5159_1_supp_876279_pc44ty.xls", and thus I was not able to tell which file is for HH24g-HH24 pair or HH24g-HH29 pair.

We apologise for this but we correctly labelled the file names so the upload process must be scrambling them. We have added a note during the upload for this to be corrected if it happens again.

iii) Description of KEGG analyses is incomplete in Supplementary files. For example, in Supplementary file 2, KEGG Pathways of NPY2R is described as "Neuroactive ligand-receptor in…".

We have manually corrected all of the pathway names.

iv) Description of KEGG pathways in Supplementary file 1, Supplementary file 2 and Figure 1E is not consistent. For example, in Supplementary file 2, KEGG Pathways of KCNQ1 is described as "Adrenergic signaling in cardio…", but in Figure 1E, it was "Adrenergic". This should be explained at least in Figure legend.

We have corrected and simplified pathway names, for instance, ‘Adrenergic signalling’ rather than ‘Adrenergic signalling in cardiomyocytes’.

v) In Figure 1E, HH24g vs. HH29 is on the left side in Up-regulated pathways, but on the right side in Down-regulated pathways. It is confusing. They should be on the same side.

We have amended this.

3) BMP and Noggin beads implantation. i) In Figure 3K, control for HH24g+Nog+8h is not appropriate. Controls should be limb buds implanted with PBS/4mM HCl (i.e. vehicle)-soaked beads (i.e. HH24g-PBS/4mM HCl+8h).

Although we labelled this HH24g-Nog+8h the control was an affigel bead soaked in PBS/4mM HCl. For brevity, we have mentioned the full details of the controls in the Materials and methods section.

ii) In Figure 3K, controls for HH20+BMP2low/higH^+^8h are not appropriate. Controls should be limb buds implanted with PBS/4mM HCl -soaked beads (i.e. HH20-PBS/4mM HCl+8h).

For our flow cytometry experiments the most accurate control is the left wing buds of the same embryos. Over several papers we have shown that proportions of G1-phase cells differ by less that 1% between equivalent tissue in left and right wing buds of the same embryos. However, we agree with the reviewer that it is important to verify if control PBS beads affect the cell cycle. Therefore, we have treated HH20 wing buds with PBS/4mM HCl^-^soaked beads for 8h and shown that this does not significantly affect cell cycle parameters compared to the equivalent tissue in the contralateral wing bud.

iii) In Figure 3L, control is not appropriate. Again, control should be limb buds implanted with PBS/4mM HCl^-^soaked beads.

This control was PBS/4mM HCl^-^soaked beads and we have mentioned this in the Materials and methods section.

4) FGF beads implantation. i) Vehicle for FGF beads implantation is not mentioned in MM.

Heparin beads were used and the recombinant FGF protein was dissolved in PBS.

ii) In Figure 4G, control should be limb buds implanted with vehicle-soaked beads.

We have included this experiment and shown that heparin/PBS beads implanted at HH27 do not up-regulate *Dusp6* after 6h.

iii) In Figure 4I, again, control should be HH27+vehicle+6h.

Following on from point 3ii, we have performed an experiment in which we implanted control heparin beads soaked in PBS to HH27 wing buds and shown that this does not significantly affect cell cycle parameters after 6h compared to the equivalent tissue in the contralateral wing bud.